# Loneliness and Relationship Well-Being: Investigating the Mediating Roles of Relationship Awareness and Distraction among Romantic Partners

**DOI:** 10.3390/bs14060439

**Published:** 2024-05-24

**Authors:** Thomas B. Sease, Emily K. Sandoz, Leo Yoke, Julie A. Swets, Cathy R. Cox

**Affiliations:** 1Institute of Behavioral Research, Texas Christian University, Fort Worth, TX 76129, USA; c.cox@tcu.edu; 2Psychology Department, University of Louisiana at Lafayette, Lafayette, LA 70504, USA; emily.sandoz@louisiana.edu; 3San Fransico Center for Compassion Focused Therapies, San Francisco, CA 94102, USA; leo.yoke@sfcompassion.com; 4Eastern Washington University at Bellevue College, Bellevue, WA 98007, USA; jswets@ewu.edu

**Keywords:** loneliness, social connection, romantic relationship, trust, conflict, well-being

## Abstract

Loneliness arises when there is a discrepancy between one’s desired and actual social connection with others. Studies examining the effects of loneliness in romantic relationships show that people who are lonely are less satisfied and committed to their romantic relationships. The present study explored the association between loneliness and romantic relationship well-being. Using a cross-sectional design, loneliness was correlated with relationship commitment, trust, and conflict. Relationship awareness, but not relationship distraction, statistically mediated the association between loneliness, relationship conflict, and relationship trust. The indirect effect of loneliness on relationship well-being was only present in people reporting low and medium levels of psychological inflexibility. Implications are discussed for acceptance- and mindfulness-based interventions for persons in romantic relationships.

## 1. Introduction

Loneliness arises when there is a discrepancy between one’s desired and actual social connection with others [1,2]. Research has shown that loneliness is a universally experienced affective state [3,4,5], with nearly 11% of all people reporting feelings of loneliness at any one time [6,7]. Loneliness differs from concepts like solitude in that people who are lonely generally crave the social connection they are lacking [8]. Failure to respond effectively to feelings of loneliness and reconnect with others can have deleterious effects, having been correlated with decreased physical and psychological well-being [9,10,11].

### 1.1. Loneliness

As a non-scientific term, the word “loneliness” is commonly used in situations that emphasize a person’s physical proximity with other people. For example, someone who spends a considerable amount of time alone may be referred to as a “loner” and thought of as a “lonely person”. This colloquial usage of the word contrasts the scientific literature, which instead uses the term loneliness to describe the experience of being psychologically isolated. This point is perhaps best exemplified by scientific definitions describing loneliness as the absence of meaningful social relationships [1,2,12,13]. Empirical studies have identified reliable precursors of loneliness, such as social anxiety, rejection sensitivity, and social rejection [14,15,16]. These data showing that loneliness can be precipitated in experimental settings suggest that loneliness is likely context-dependent rather than context-independent [17,18]. For example, people commonly report feeling lonely following the loss of a loved one or the dissolution of a close relationship.

Extant literature also suggests that loneliness is associated with patterns of behavior across various levels of analysis [19,20,21,22]. For example, people who are lonely show more cardiovascular activation, a risk factor for chronic heart disease, than people who are not lonely [23]. Feelings of loneliness are a robust predictor of psychopathology [24,25,26] and have been positively correlated with social avoidance and withdrawal in young adults [27]. Collectively, interpreting these findings suggest loneliness is a response class of behaviors precipitated in situations where we lack social connection.

### 1.2. Loneliness in Romantic Relationships

Several theorists have noted humans’ fundamental need to belong and their desire for meaningful social relationships [28,29,30,31] Romantic relationships have specifically been identified as having a unique effect on health; for example, married persons report feeling more supported, having less stress, and experiencing greater well-being as compared to their single counterparts [32,33]. Similar results have also been shown among relationship partners prior to marriage [34]. According to Myers [35], the associative link between relationship status and well-being is not merely explained by relationship status alone, but that the quality of the partnership matters. For instance, higher relationship quality (e.g., satisfaction, commitment, trust) is associated with increased levels of happiness, life satisfaction, positive affect, and self-esteem [34,36,37]. Conversely, lower relationship quality (e.g., higher conflict, less trust, more relationship anxiety) is related to increased levels of anxiety, depression, aggression, and substance abuse, as well as poorer immune and endocrine functioning [38,39,40,41,42].

There exist a handful of studies that have examined the association between loneliness and relationship well-being. These studies have mostly investigated partner satisfaction and commitment as a predictor of loneliness [43,44,45,46]. Certainly, these associations can be interpreted bidirectionally as people who are lonely find their relationships less satisfying. This alternative interpretation of the data may be especially relevant for people experiencing chronic feelings of loneliness. Said differently, the interpersonal consequences of loneliness may be especially salient for people with “trait-based loneliness”—that is, loneliness that is present for a person in general across many different contexts. An 8-year longitudinal study including heterosexual dyads in a romantic relationship showed that loneliness prospectively predicted one’s own relationship dissatisfaction and their partner’s relationship dissatisfaction [47]. As such, a potential next step to better understanding the association between loneliness and romantic relationship well-being is to identify mechanisms participating in this relationship, so that clinical interventions can be tailored to redress the negative interpersonal outcomes connected to loneliness.

### 1.3. Potential Mechanisms: Relationship Awareness and Distraction

Mindfulness is one concept that could be involved in the relationship between loneliness and relationship well-being [48]. That is, mindfulness has been shown to confer several interpersonal benefits [49,50,51] and may therefore impact how loneliness affects people in romantic relationships. The term mindfulness has been broadly defined as a behavioral process including (1) an acute awareness of thoughts and feelings as being separate from reality, (2) the ability to remain present with intrusive thoughts without avoidance, (3) a predisposition towards accepting negative internal experiences, and (4) letting go of aversive thoughts without becoming preoccupied with their content [52]. In practice, mindfulness is a purported mechanism of change among many third-wave psychotherapies, yielding moderate to large effect sizes in youth [53], older adults [54], people with depression and anxiety [55], disordered eating [56], and psychosis [57]. Mindfulness-based interventions foster therapeutic change by deemphasizing the content of subtle events and building skills in areas related to emotion regulation, psychological flexibility, and compassion.

Additionally, relationship awareness, or the tendency to remain in the present moment and actively engaged with one’s romantic partnership [58,59,60], and relationship distraction, or being continually out of touch or neglectful of one’s romantic relationship [61], could be mindfulness-based processes mediating the association between loneliness and romantic relationship well-being. Longitudinal evidence has shown relationship awareness and distraction are differentially associated with positive (e.g., satisfaction, dedication, affection) and negative (e.g., relationship anxiety, conflict, insensitivity) relationship outcomes [61]. Conceptually, loneliness may decrease peoples’ sensitivity to cues of social reinforcement and increase their sensitivity to cues of social punishment. Indeed, people who are lonely are less likely to detect commitment cues (i.e., receiving words of affirmation from a close friend or romantic partner) when compared to their non-lonely counterparts [62] and show an increased sensitivity to romantic relationship threat [63]. Taken together, the current study investigated relationship awareness and relationship distraction as potential mediators of the relationship between loneliness and romantic relationship well-being.

Another variable that could impact the associative link between loneliness, relationship awareness, relationship distraction, and romantic relationship well-being is psychological inflexibility. Psychological inflexibility has been conceptualized as the “rigid dominance of psychological reactions over chosen values and contingencies in guiding action” [64]. Being more psychologically inflexible, as compared to more psychologically flexible, has deleterious effects, such as increased anxiety and depression [65]. Psychological inflexibility also negatively impacts romantic relationship well-being. In a recent meta-analysis, psychological inflexibility was correlated with lower relationship satisfaction, lower sexual satisfaction, higher conflict, and more physical aggression [66]. Thus, someone that is more psychologically flexible may be able to overcome the negative consequences associated with loneliness in romantic relationships. In other words, psychological inflexibility may moderate the anticipated indirect effect of loneliness on decreased romantic relationship well-being through relationship awareness and relationship distraction.

### 1.4. Current Study

Studies focusing on loneliness in the context of romantic relationships have demonstrated that loneliness can be precipitated by feelings of relationship dissatisfaction [43,44,45,46]. However, one question that remains is how loneliness may lead someone to experience interpersonal difficulties in close relationships. Uncovering manipulable processes supporting the association between loneliness and romantic relationship well-being could afford therapists additional clinical targets with the potential to improve client outcomes.

The purpose of this study was to identify a pathway through which loneliness contributes to decreased relationship well-being among people in a romantic relationship. The authors hypothesized that loneliness would be negatively associated with relationship awareness and positively associated with relationship distraction. It was also expected that relationship awareness and relationship distraction would simultaneously mediate the association between loneliness and relationship well-being. Finally, a first stage moderated mediation model investigated psychological inflexibility as a moderator of the indirect effect loneliness had on relationship well-being. Although the moderated-mediation model was deemed exploratory in nature, we expected the indirect effect of loneliness on romantic relationship outcomes to only be significant at high levels of psychological flexibility. In other words, we did not expect loneliness to decrease relationship well-being for people reporting low levels of psychological inflexibility.

## 2. Method

### 2.1. Participants

Two hundred and twenty people showed an initial interest in the study by starting the survey and 210 people completed more than 75% of the study survey. After removing data from participants who (1) missed an attention check item, (2) did not provide consent for us to use their data, and (3) reported that they were not currently in a romantic relationship, the final sample consisted of 201 people ranging in age from 19–72 (*M* = 37.53, *SD* = 11.70). As illustrated in Table 1, most of the people in this study were women (*n* = 138, 68.7%), White (*n* = 165, 82.0%), heterosexual (*n* = 170, 84.6%), and married (*n* = 112, 55.7%).

### 2.2. Procedure

Two hundred and ten people were recruited from Cloud Research [67], an online platform that provides academic researchers with high quality data by prescreening workers on Amazon’s Mechanical Turk (MTurk). Inclusion criteria required respondents to be (1) at least 18 years old, (2) fluent in the English language, (3) live in the United States, and (4) be in a romantic relationship. Study participation was restricted to people who have completed between 0–500 surveys on MTurk with an approval rating greater than or equal to 0.95. All respondents were compensated $1 for participating in this study.

The study survey was created in Qualtrics with a hyperlink posted online through Cloud Research. Respondents were presented with a vague title (i.e., “Research Study for People in a Romantic Relationship”) and a description of the study that included its purpose and inclusion criteria. People who were interested in the study could click on the survey link, which connected participants to the informed consent page. Participants who provide informed consent could proceed with the remainder of the survey, which was composed of counterbalanced instruments of loneliness, attentive awareness, relationship distraction, relationship trust, relationship conflict, relationship commitment, and demographic questions. Four attention checks presented on a Likert scale (e.g., “Select *Strongly Disagree* for this item”) were randomly incorporated throughout the survey. One open-ended question requiring participants to type “I am not a robot” in a text blank was also included. People who missed any of the five attention checks were excluded from data analysis. Similarly, participants who provided information during the study that would make them ineligible for the project (e.g., not being in a romantic relationship) were removed from the dataset. Missing attention checks and providing information that would make someone ineligible for the study did not affect participants’ right to compensation. This study was pre-registered prior to the start of data collection and all materials associated with this study are available at https://osf.io/u4sc8/?view_only=9215b02fccd04bf0ad5a9794ef034e3e (accessed on 17 July 2022). Data collection for this project was completed in August 2022, and this study was approved by the first author’s Institutional Review Board prior to data collection.

### 2.3. Measures

#### 2.3.1. Demographics

A brief demographics form was administered at the end of the study to capture respondents’ age, gender identity, sexuality, and race. This form also included questions asking about participants’ romantic relationships (e.g., length, dating vs. married, etc.).

#### 2.3.2. Relationship Conflict

The 6-item Relationship Conflict Scale [68] was presented on a 7-point Likert scale (1 = *Strongly Disagree*, 7 = *Strongly Agree*) to measure the amount of conflict in respondents’ romantic relationships. Sample items include: “There is a lot of conflict in my relationship”, “My partner and I have a lot of conflicts”, and “My partner and I are always in agreement on major issues” (reversed). The Relationship Conflict Scale has been tested among people in romantic relationships, showing good internal reliability (α = 0.83) and validity [68]. Scores for relationship conflict were calculated by taking the mean of all items, with higher scores indicating more conflict.

#### 2.3.3. Relationship Commitment

Relationship commitment was assessed using the commitment scale on the Investment Model Scale [69]. This instrument was presented on a 9-point Likert scale (0 = *Do not Agree at All*, 8 = *Agree Completely*) and measured relationship commitment using items such as, “I am committed to maintaining my relationship with my partner”, “I am oriented toward the long-term future of my relationship”, and “It is likely I will date someone other than my partner in the next year” (reversed). The relationship commitment subscale has shown strong psychometric properties in past research [69] and was scored by taking the mean of all items.

#### 2.3.4. Relationship Trust

Relationship trust was measured using the 17-item Trust in Close Relationships Scale [70]. Items for this scale were presented on a 7-point Likert scale (−3 = *Strongly Disagree*, 3 = *Strongly Agree*) and instructed respondents to answer each question as it pertains to their romantic partner. Sample items include: “My partner has proven to be trustworthy, and I am willing to let him/her engage in activities which other partners find too threatening”, “I have found my partner is usually dependable”, and “My partner is very unpredictable. I never know how he/she is going to act from one day to the next” (reversed). Items that used gendered language, such as “Even when my partner makes excuses which sound rather unlikely, I am confident that he/she is telling the truth”. were changed to use they/them pronouns to describe their partner. The Trust in Close Relationships Scale has demonstrated an acceptable internal reliability score (α = 0.70) and validity [70]. Scores for relationship trust were computed by taking the mean of all items.

#### 2.3.5. Loneliness

Loneliness was assessed using the well-established UCLA Loneliness scale [71,72]. This 20-item instrument was presented on a 4-point Likert scale (0 = *I never feel this way*, 3 = *I often feel this way*) and asked respondents to rate how often they experience feelings of social isolation. Example items include: “I am unable to reach out and communicate with those around me”, “I find myself waiting for people to call or write”, and “My interests and ideas are not shared by those around me”. Studies evaluating the psychometric properties of the UCLA Loneliness Scale have revealed this instrument is both reliable and valid [73,74]. For people in romantic relationships, the UCLA Loneliness Scale has shown an internal reliability score of 0.93 [75]. Loneliness scores were computed by taking the mean of all items, with higher scores indicating more loneliness.

#### 2.3.6. Relationship Awareness and Distraction

Relationship awareness and distraction were measured using the newly developed Attentive Awareness in Relationships Scale [61]. The AAIRS is a 16-item instrument presented on a 7-point Likert scale (1 = *Strongly Disagree*, 7 = *Strongly Agree*). Respondents are instructed to answer each item as it pertains to respondents’ sensitivity to the needs of their romantic relationship. Sample items include: “I was in touch with the ebb and flow of feelings in my romantic relationship” and “I was distracted and did not pay much attention to my romantic relationship” for relationship awareness and distraction, respectively. The AAIRS is an item response theory optimized instrument, demonstrating acceptable internal consistency (α = 0.87 for relationship awareness; α = 0.93 for relationship distraction), convergent, divergent, and predictive validity [61]. Relationship awareness and relationship distraction scores were calculated by taking the mean of all items within their respective scale.

#### 2.3.7. Psychological Inflexibility

Psychological inflexibility was measured using the Multidimensional Psychological Flexibility Inventory [75]. This study used the shortened 12-item psychological inflexibility scale that is presented on a 7-point Likert scale (1 = *Strongly Disagree*, 7 = *Strongly Agree*). Sample items include: “When I had a bad memory, I tried to distract myself to make it go away”, “I thought some of my emotions were bad or inappropriate and I shouldn’t feel them”, and “I criticized myself for having irrational or inappropriate emotions”. The MPFI is an item response theory optimized instrument that has shown acceptable internal reliability (α = 0.88–0.92), convergent, content, and divergent validity [75]. The MPFI was scored by taking the mean of all items, with higher scores meaning more psychological inflexibility.

#### 2.3.8. Analytic Plan

Using R Studio [76], descriptive statistics and regression diagnostics were examined for all variables of interest. Pearson’s product-moment correlation investigated the associations among loneliness, relationship awareness, distraction, and measures of relationship well-being. To test the study’s main hypothesis, three parallel mediation models were fitted to the data with loneliness predicting each indicator of relationship well-being through the proposed mediators (i.e., mindful awareness, mindful distraction). The indirect effect of loneliness on relationship well-being was estimated by simultaneously taking the product of the *a*_1_ and *a*_2_ path and the *b*_1_ and *b*_2_ path [77]. These effects included 5000 bootstrap reiterations with a 95% confidence interval [78]. A Monte Carlo power analysis [79] including 20,000 replications suggested that between 270 and 277 people were required to observe a significant indirect effect through each mediator—assuming a large effect size (*r* = 0.60) between the mediators and a moderate effect size (*r* = 0.40) among all other variables. With a final sample of 201, the power analysis revealed that this study was underpowered.

Next, a pre-registered exploratory analysis tested whether psychological inflexibility moderated the association between loneliness and mindful awareness using hierarchical multiple linear regression. Simple slope analysis [80] was used to decompose the 2-way interaction between loneliness and mindful awareness at low (1 *SD* below the mean), medium (at the mean), and high (1 *SD* above the mean) levels of psychological inflexibility. Non-significant predictors were removed from the mediation model and then combined with the moderation results in a first-stage moderated mediation model. A critical value of 0.05 determined statistical significance and the data for this study were not analyzed until all participant responses were collected.

## 3. Results

Descriptive statistics and internal reliability scores for all measures are displayed in Table 2. Correlation analyses showed that all measures were significantly associated in a theoretically consistent direction.

To test the authors’ main hypothesis, three parallel mediation models investigated whether relationship awareness and relationship distraction simultaneously mediated the association between loneliness and measures of relationship well-being (see Figure 1). Loneliness was significantly associated with relationship commitment (*b* = −0.43, *SE* = 0.12, *t* = 3.59, *p* < 0.001), trust (*b* = −0.76, *SE* = 0.11, *t* = 7.04, *p* < 0.001), and conflict (*b* = 0.84, *SE* = 0.13, *t* = 6.42, *p* < 0.001). More importantly, loneliness was negatively associated with relationship awareness (*b* = −0.37, *SE* = 0.09, *t* = 4.43, *p* < 0.001; *a*_1_ path) and positively associated with relationship distraction (*b* = 0.40, *SE* = 0.08, *t* = 5.25, *p* < 0.001; *a*_2_ path). While controlling for other variables in the model, relationship awareness was associated with relationship trust (*b* = 0.50, *SE* = 0.12, *t* = 4.16, *p* < 0.001; *b*_1_ path) whereas the association between relationship distraction and relationship trust did not reach statistical significance (*b* = −0.12, *SE* = 0.13, *t* = 0.95, *p* = 0.343; *b*_2_ path). The indirect effect of loneliness on relationship trust through relationship awareness was significant, 95% *C.I.* [−0.32, −0.08], suggesting relationship awareness statistically mediated the relationship between loneliness and relationship trust.

Similar results were observed when relationship conflict was entered into the regression model as the dependent variable. When holding other variables in the model constant, relationship awareness was negatively associated with relationship conflict (*b* = −0.41, *SE* = 0.15, *t* = 2.71, *p* < 0.001; *b*_1_ path) and relationship distraction not significantly related to relationship conflict (*b* = 0.21, *SE* = 0.17, *t* = 1.28, *p* = 0.201; *b*_2_ path). The indirect effect including 5000 bootstrap reiterations and a 95% confidence interval for relationship awareness was significant, 95% *C.I.* [0.04, 0.31]. In contrast, when relationship commitment was examined as the outcome variable, relationship distraction was correlated with relationship commitment (*b* = −0.33, *SE* = 0.15, *t* = 2.20, *p* = 0.029; *b*_1_ path) but not relationship awareness (*b* = 0.25, *SE* = 0.14, *t* = 1.82, *p* = 0.07; *b*_2_ path). The indirect effect, however, was non-significant for both relationship distraction, 95% *C.I.* [−0.31,0.01] and awareness, 95% *C.I.* [−0.23,0.03], meaning there was no mediational effect.

As an exploratory analysis, a hierarchical multiple linear regression tested whether psychological inflexibility moderated the association between loneliness and relationship awareness (see Table 3). At Step 1, the main effects of loneliness (centered) and psychological inflexibility (centered) were entered into the regression model as predictors of relationship awareness. Then, the interaction between loneliness and psychological inflexibility was entered into the model as a predictor variable, while controlling for their main effects. Results showed that the interaction between loneliness and psychological inflexibility was significant when predicting relationship awareness (*b* = 0.21, *SE* = 0.07, *t* = 2.82, *p* = 0.005). Simple slope analysis showed that there was a negative association between loneliness and relationship awareness at low (*b* = −0.54, *SE* = 0.14, *t* = 3.79, *p* < 0.001) and medium (*b* = −0.33, *SE* = 0.11, *t* = 3.12, *p* < 0.001) levels of psychological inflexibility (see Figure 2). In contrast, the association between loneliness and relationship awareness was non-significant at high levels of psychological inflexibility (*b* = −0.11, *SE* = 0.11, *t* = 1.00, *p* = 0.320).

In extension of the results above, a first stage moderated mediation model examined psychological inflexibility as a moderator of the mediational effect relationship awareness had on the relationship between loneliness and measures of relationship well-being (see Figure 3 and Figure 4). Results showed that the indirect effect of loneliness on relationship trust through relationship awareness was significant at low, 95% *C.I.* [−0.50, −0.15], and medium, 95% *C.I.* [−0.32, −0.07], levels of psychological inflexibility. Conversely, at high levels of psychological inflexibility, the indirect effect was non-significant, 95% *C.I.* [−0.20, 0.08]. Likewise, the mediational effect of relationship awareness on the relation between loneliness and relationship conflict was only present at low, 95% *C.I.* [0.13, 0.50], and medium levels, 95% *C.I.* [0.06, 0.32], of psychological inflexibly, but not high, 95% *C.I.* [−0.08, 0.20].

## 4. Discussion

Loneliness is an affective state ubiquitous to the human condition that contributes to poor physical and psychological health [9,10,11]. In this study, we investigated whether relationship awareness and relationship distraction—two processes of mindfulness relevant in romantic relationships—statistically mediated the association between loneliness and measures of romantic relationship well-being. We found that relationship awareness, but not relationship distraction, had a mediational effect on the relationship between loneliness and relationship conflict as well as the relationship between loneliness and relationship trust. More specifically, people who reported feeling lonely were more likely to report less awareness in their romantic relationships. This decreased attentiveness in persons’ romantic relationships was correlated with more self-reported conflict and less trust of their partner.

Further, we tested whether psychological inflexibility moderated these mediational results. A significant two-way interaction between loneliness and psychological inflexibility was observed; loneliness was only correlated with less relationship awareness at low and medium levels of psychological inflexibility. Upon further investigation, the association between psychological inflexibility and relationship awareness was significant, even while controlling for the effects of loneliness. It thus appears that, in the current study, the effect of loneliness on relationship awareness did not matter if the person was also reporting high levels of psychological inflexibility. These results converge with extant literature showing the negative effects psychological inflexibility has on romantic relationships [see 66 for a full review], and suggests that the practical implications derived from the mediational effect observed in this study may be most relevant for people with low levels of psychological inflexibility.

These results have direct implications for mindful- and acceptance-based interventions, such as Acceptance and Commitment Therapy [81]. Loneliness is a common psychological concern for people presenting with depression, anxiety disorders, schizophrenia, and suicidal ideation [15,25,26], and is associated with less positive treatment outcomes [82]. Thus, the information gathered from this study can be used to identify and target processes in psychotherapy that contribute to poor psychosocial functioning in several clinical populations. For clients in romantic relationships, this study identified a manipulable process (i.e., relationship awareness) that statistically mediated the relationship between loneliness and measures of romantic relationship well-being. Therapists working with romantic partners may aim to increase clients’ attentiveness to their partner to bolster positive relationship outcomes. Indeed, interventions using mindfulness as the purported mechanism of change have shown preliminary effectiveness in improving relationship acceptance and satisfaction [83,84,85], suggesting techniques focusing on mindful awareness may show similar benefits.

### Limitations and Future Directions

This study used a cross-sectional design to investigate the role that relationship awareness and distraction had on the association between loneliness and relationship well-being. This eliminates the possibility of inferring causation from the proposed mediation model, and thus, a temporal relationship among these variables cannot be established. Succeeding studies will need to replicate this project’s findings longitudinally to conclude the directionality of these associations. It may also be useful to explore other mediators that may be involved in explaining the relationships between loneliness and interpersonal outcomes in romantic relationships. In other words, the meditation models in this study only explained 36.5% and 27.8% of the observed variance in relationship trust and conflict, respectively. Thus, forthcoming work could focus on other behavioral characteristics associated with loneliness that may contribute negative interpersonal outcomes.

Second, given this study’s relatively small sample size, our design was unpowered, and it would be useful to replicate these findings using a larger, more representative sample. People in couples’ therapy could be recruited and asked to complete assessments before every session to test whether fluctuations in loneliness affect relationship instability in the manner proposed here. Ideally, these investigations would include responses from both partners to assess how one partner’s feelings of loneliness affect their partner’s behavior in a manner that could be detrimental for the relationship. For example, Actor Partner Interdependence Modeling [86] allows researchers to statistically control for the dependence between romantic partners and create models that can be used to determine how one partner’s behavior affects their own and their partner’s behavior.

Finally, how relationship well-being was operationalized is perhaps the largest limitation of this study. Relationship well-being was conceptualized as romantic relationships low in commitment, low in trust, and high in conflict. Although these variables have been repeatedly correlated with poor relationship outcomes among romantic partners [87,88,89,90], there are several variables that could have been justifiably included as a proxy for relationship well-being (e.g., relationship satisfaction, infidelity, affection). Along with these measures being self-report assessments, future work in the area would benefit from using behavioral assessments of relationship instability, including the number of conflicts per week, infidelity, or relationship dissolution [91]. This would provide more tangible empirical evidence demonstrating that decreased relationship awareness in response to loneliness is determinantal for people in romantic relationships.

## 5. Conclusions

This project investigated the roles relationship awareness and relationship distraction played on the association between loneliness and romantic relationship well-being. The results showed that loneliness was associated with less self-reported relationship awareness, which in turn was associated with more relationship conflict and less relationship trust. Psychological inflexibility was also found to moderate the association between loneliness and relationship awareness. Specifically, the indirect effect of loneliness on relationship outcomes through relationship awareness was only present at low- to medium-levels of psychological inflexibility. These data provide insight into how loneliness may contribute to worsened interpersonal outcomes in romantic relationships. As such, this study provides useful information that can be incorporated into the curriculum of clinical programs designed to redress feelings of isolation for people in romantic relationships.

## Figures and Tables

**Figure 1 behavsci-14-00439-f001:**
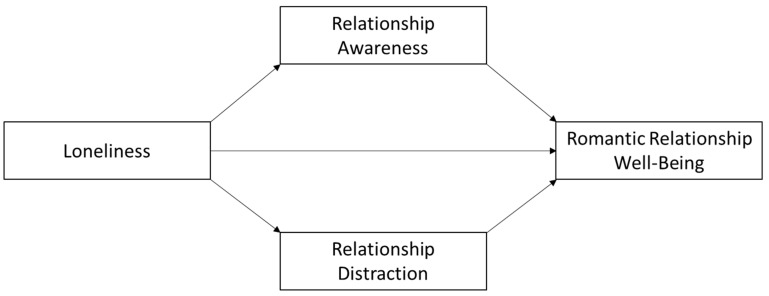
Theorized Path Model.

**Figure 2 behavsci-14-00439-f002:**
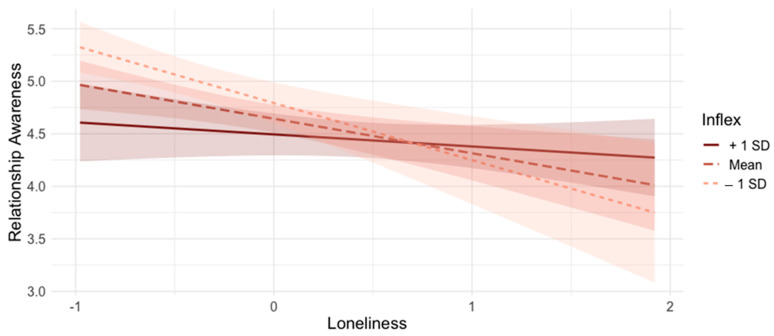
Two-way Interaction between Loneliness and Psychological Inflexibility; Note. Inflex = Psychological Inflexibility.

**Figure 3 behavsci-14-00439-f003:**
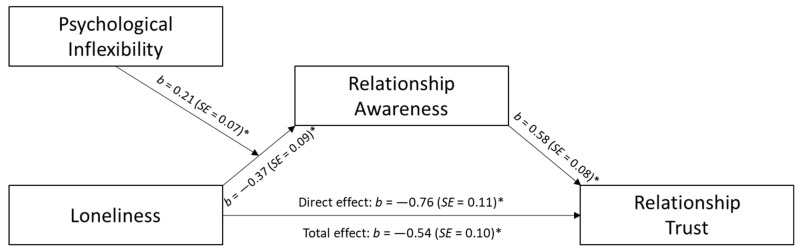
Predicting Relationship Trust; Note. * *p* < 0.01.

**Figure 4 behavsci-14-00439-f004:**
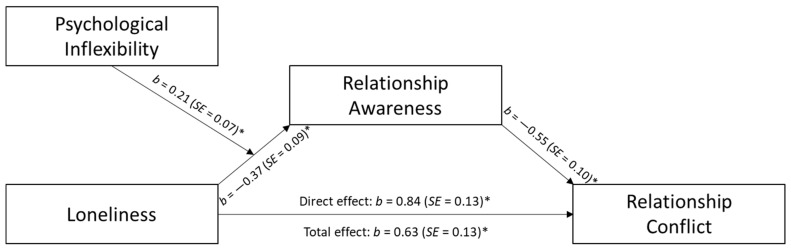
Predicting Relationship Conflict. Note. * *p* < 0.01.

**Table 1 behavsci-14-00439-t001:** Descriptive Statistics.

Characteristic	*n* (%)
Gender Identity	
Woman	138 (68.7%)
Man	60 (30.0%)
Non-Binary or Transgender	3 (0.03%)
Race	
White	165 (82.1%)
BIPOC	36 (17.9%)
Sexual Orientation	
Heterosexual	170 (84.6%)
Gay or Lesbian	5 (2.5%)
Bisexual	21 (10.4%)
Another Sexuality	5 (2.5%)
Relationship Status	
Married	112 (55.6%)
Engaged	13 (6.5%)
Domestic Partnership	14 (7.0%)
Committed Relationship	57 (28.4%)
Casually dating someone	5 (2.5%)

Note. BIPOC = Black, Indigenous, or People of Color.

**Table 2 behavsci-14-00439-t002:** Descriptive Statistics and Bivariate Correlations.

Scale	Mean (*SD*)	1	2	3	4	5	6	7
1. Loneliness	0.98 (0.72)	0.96						
2. Inflexibility	3.02 (1.03)	0.59	0.93					
3. Awareness	4.73 (0.91)	−0.30	−0.28	0.93				
4. Distraction	2.11 (0.84)	0.35	0.35	−0.76	0.90			
5. Trust	5.44 (1.23)	−0.45	−0.47	0.52	−0.47	0.94		
6. Conflict	2.63 (1.47)	0.41	0.44	−0.43	0.42	−0.82	0.92	
7. Commitment	8.25 (1.52)	−0.25	−0.31	0.38	−0.40	0.56	−0.46	0.87

Note. All correlations were statistically significant at a critical *p* value of 0.05. Internal reliability scores (α) are displayed on the diagonal.

**Table 3 behavsci-14-00439-t003:** Moderation Analysis.

	*B*	*Β*	*SE*	*t*	Sig.	*R* ^2^
Step 1						10.6%
Loneliness	−0.26	−0.21	0.10	2.50	<0.001	
Inflexibility	−0.14	−0.16	0.07	1.89	0.060	
Step 2						14.0%
Loneliness	−0.33	−0.26	0.11	3.12	<0.001	
Inflexibility	−0.14	−0.16	0.07	2.02	0.045	
Interaction	0.21	0.17	0.07	2.82	0.005	

Note. Interaction = Loneliness × Psychological Inflexibility.

## Data Availability

Data will be made available upon request by the corresponding author.

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
