# Peer review of "Loneliness and Relationship Well-Being: Investigating the Mediating Roles of Relationship Awareness and Distraction among Romantic Partners"

_behavsci, 2024, doi:10.3390/bs14060439_

Round 1
Reviewer 1 Report
Comments and Suggestions for Authors
Thank you for the opportunity to read this highly interesting article. The study explored the relationship between loneliness and romantic relationship well-being. Previous studies have shown that there is a connection between the experience of loneliness and satisfaction in romantic relationship. The study is quantitative, and data consist of 201 American respondents.
All in all, the article is well written. Especially method is described carefully, and it seems that analysis is exact. Authors are familiar with previous studies of the topic. However, I hope the authors could rethink and improve the following points:
- Chapter 1.3. When reading the text, I was surprised you had chosen to write about mindfulness. While I am very familiar with different mindfulness techniques, I would absolutely see better arguments why you chose it. I am sure, mindfulness may be a useful concept for your study, but please, write more about it both in chapter 1.3. and in 4.
- Chapter 3. Tables and figures are good but Figures 3 and 4 are grainy.
- Chapter 4. Please, clarify what is the relevance of your study. What is new compared to previous studies? What are the limitations of respondents? ( I think data is quite small.) Highlight the importance of this article. Even though the theme is very interesting, I didn´t find very much new knowledge, and I think the results were predictable.
Although the article is not very ambitious, it is carefully prepared and suitable for the journal.
Author Response
Reviewer #1
All in all, the article is well written. [The] method is described carefully, and it seems that analysis is exact. Authors are familiar with previous studies of the topic. However, I hope the authors could rethink and improve the following points:
- Chapter 1.3. When reading the text, I was surprised you had chosen to write about mindfulness. While I am very familiar with different mindfulness techniques, I would absolutely see better arguments why you chose it. I am sure mindfulness may be a useful concept for your study, but please, write more about it both in chapter 1.3. and in 4.
- Our choice to focus on mindfulness stemmed from work on the potential interpersonal benefits mindfulness may confer. We agree that this point was not clear enough in the original paper, and thus edited and added a few sentences in the literature review so that this is more apparent (i.e., Chapter 1.3, Lines 91-95). Additionally, we have incorporated citations into this section so that readers may familiarize themselves with literature on this topic.
- Chapter 3. Tables and figures are good but Figures 3 and 4 are grainy.
- Thank you for drawing our attention to this. We recreated the images to appear more clearly in the revised manuscript.
- Chapter 4. Please, clarify what is the relevance of your study. What is new compared to previous studies? What are the limitations of respondents? (I think data is quite small.) Highlight the importance of this article. Even though the theme is very interesting, I didn´t find very much new knowledge, and I think the results were predictable.
- We have added to the discussion section on how this study differs from other research in this area, as well as, accentuating how this study adds to our understanding of loneliness, mindfulness, and romantic relationships. We have additionally added to the limitations section about the limited sample size and how this may affect the interpretation of our study’s results.
Reviewer 2 Report
Comments and Suggestions for Authors
This is a strong article, which raises significant issues on loneliness, providing important findings on its causes and consequent methods of coping with it. The authors present extensive knowledge of the relevant literature on issues central in their study, and the article is written in a clear and persuasive manner.
Since I am not a psychologist, I cannot comment on the methodological aspects of the paper, and instead, have focussed on the conceptual ones.
The authors rightly claim that loneliness expresses the “absence of meaningful social relationships.” However, they also argue that “loneliness is an emotion.” I disagree. Although loneliness is a type of affective attitude, it is not an emotion. Emotions have an object at which they are directed, and loneliness does not have such an object. In this sense, loneliness is closer to a kind of mood, which is a background state of mind, relating to fundamental ways in which one is open to and relates to the world (Spiegel, 2023). An important finding of the study is that “loneliness was only correlated with less relationship awareness at low and medium levels of psychological inflexibility.” The lower level of flexibility is indeed relevant to the claim that the single most commonly proven skill of importance to our mental health and emotional wellbeing is psychological flexibility (Hayes et al., 2022). The absence of this considerably reduces psychological richness (Oishi & Westgate, 2021). The authors rightly indicate the importance of awareness in loneliness. Is loneliness also associated with information avoidance? It would be interesting for this to be explored further.
Author Response
This is a strong article, which raises significant issues on loneliness, providing important findings on its causes and consequent methods of coping with it. The authors present extensive knowledge of the relevant literature on issues central in their study, and the article is written in a clear and persuasive manner. Since I am not a psychologist, I cannot comment on the methodological aspects of the paper, and instead, have focused on the conceptual ones.
- The authors rightly claim that loneliness expresses the “absence of meaningful social relationships.” However, they also argue that “loneliness is an emotion.” I disagree. Although loneliness is a type of affective attitude, it is not an emotion. Emotions have an object at which they are directed, and loneliness does not have such an object. In this sense, loneliness is closer to a kind of mood, which is a background state of mind, relating to fundamental ways in which one is open to and relates to the world (Spiegel, 2023). An important finding of the study is that “loneliness was only correlated with less relationship awareness at low and medium levels of psychological inflexibility.” The lower level of flexibility is indeed relevant to the claim that the single most commonly proven skill of importance to our mental health and emotional wellbeing is psychological flexibility (Hayes et al., 2022). The absence of this considerably reduces psychological richness (Oishi & Westgate, 2021). The authors rightly indicate the importance of awareness in loneliness. Is loneliness also associated with information avoidance? It would be interesting for this to be explored further.
- We appreciate Reviewer 2’s thoughtful response related to our conceptualization of loneliness in the current paper. While we believe a thorough discussion on this distinction is important, it is beyond the scope of the paper. We have edited the entire manuscript to refer to loneliness as an affective state rather than an emotion.
Reviewer 3 Report
Comments and Suggestions for Authors
The authors provide a conceptual distinction between loneliness and solitude, and focus on the predictive role that loneliness might play in romantic relationships. Though this study has practical significance and the data analysis methods seem sound, the following issues need to be concerned and improved for authors.
Firstly, the study lacks sample size calculation before recruiting participants. If the authors believe the sample size is sufficient or does not plan to increase it further, they could conduct a post-hoc power analysis to inform readers of the statistical power that the study’s sample size achieved.
Secondly, in this article, relationship conflict is examined as an aspect of relationship well-being, but it seems to not be mentioned at all in the introduction. The author acknowledges in the discussion that how relationship well-being was operationalized is perhaps the largest limitation of this study. In this case, the author may consider adding past literature on whether there have been any findings between loneliness and relationship conflict, or between mindfulness-based processes and relationship conflict, in the introduction section.
Thirdly, although the investigation of the moderating role of psychological flexibility is an exploratory analysis, why the variable of “psychological inflexibility” is included in this article? The logic behind this model should be mentioned in the introduction. Moreover, in the discussion section of this article authors still do not delve into the significance of psychological inflexibility. Consequently, much elaboration and explanation still need to be added on what psychological inflexibility is and why it produces corresponding effects.
Finally, as a cross-sectional design the theoretical and practical value of this study seem to be much limited.
Author Response
Reviewer #3
The authors provide a conceptual distinction between loneliness and solitude and focus on the predictive role that loneliness might play in romantic relationships. Though this study has practical significance, and the data analysis methods seem sound, the following issues need to be concerned and improved for authors.
- Firstly, the study lacks sample size calculation before recruiting participants. If the authors believe the sample size is sufficient or does not plan to increase it further, they could conduct a post-hoc power analysis to inform readers of the statistical power that the study’s sample size achieved.
- As Reviewer 3 suggested, we now include a Monte Carlo power analysis for our proposed mediation model in the analytic plan (Chapter 2.3.8, Lines 282-287). The results of this power analysis suggested that our study was underpowered, which we have explicitly mentioned in the text and the limitations section (Chapter 4, Lines 430-432).
- Secondly, in this article, relationship conflict is examined as an aspect of relationship well-being, but it seems to not be mentioned at all in the introduction. The author acknowledges in the discussion that how relationship well-being was operationalized is perhaps the largest limitation of this study. In this case, the author may consider adding past literature on whether there have been any findings between loneliness and relationship conflict, or between mindfulness-based processes and relationship conflict, in the introduction section.
- To address Reviewer 3’s comment, we have added to our discussion on relationship conflict (Chapter 1.2, Lines 67-72) and how it is associated with worse interpersonal outcomes for people in romantic partnerships. Additionally, as suggested, we have added a sentence explaining how relationship awareness and distraction are related to the variables we used to operationalize romantic relationship well-being (Chapter 1.2, Lines 112-114).
- Thirdly, although the investigation of the moderating role of psychological flexibility is an exploratory analysis, why the variable of “psychological inflexibility” is included in this article? The logic behind this model should be mentioned in the introduction. Moreover, in the discussion section of this article authors still do not delve into the significance of psychological inflexibility. Consequently, much elaboration and explanation still need to be added on what psychological inflexibility is and why it produces corresponding effects.
- After reviewing the original manuscript, we understand how the lack of information on psychological flexibility may confuse the reader. We have addressed Reviewer 3’s concerns in two ways. First, the Introduction has been revised to include a paragraph on psychological flexibility, its correlates, and our rationale for why it may have an impact on the association between loneliness and romantic relationship well-being (see Chapter 1.3, Lines 123-136). Second, a paragraph has been added to the discussion section on psychological flexibility as a moderator in the current results (see Chapter 4, Lines 390-401).
- Finally, as a cross-sectional design the theoretical and practical value of this study seem to be much limited.
- We agree with Reviewer 3 that a cross-sectional (vs. longitudinal) design is a limitation to the current research. We have clarified this point in the revised Discussion section of the manuscript (Chapter 4, Lines 418-423).
Reviewer 4 Report
Comments and Suggestions for Authors
This is an interesting article and a good utilization of the model you described. Make a few minor edits in the literature review, though.
Comments on the Quality of English LanguageEnglish quality is good.
Author Response
Reviewer #4
- This is an interesting article and a good utilization of the model you described. Make a few minor edits in the literature review, though.
- We would like to thank Reviewer 4 for their feedback and timely review of our paper. We have addressed the minor changes Reviewer 4 provided.
Round 2
Reviewer 3 Report
Comments and Suggestions for Authors
In the revised manuscript some improvements have been made by the authors. However, I do not think the current one would be qualified to be published.
In terms of calculating the sample size, the current sample size appears to be clearly inadequate according to the information provided by the authors. Most importantly, as a cross-sectional design this article could not provide solid evidence to support the "real" relationships among variables in the model or to provide a convincing theoretical framework behind the model. The theoretical and practical contributions of this manuscript are limited.
Author Response
In reading over the reviewer's comments, we are not sure there is anything that we can do to the manuscript to satisfy the reviewer. In other words, at this point in the review process, we are unable to recruit more participants or change our research design.
We would like to point out, however, that while our proposed path model was underpower, our final models, including a mediation model and moderation analysis, are sufficiently powered. This is an important consideration given that these are the models that we discuss throughout the discussion section and that we would argue do have something to offer the field.
Moreover, while cross-sectional studies are limited, the majority of studies in psychology have been cross-sectional and a research design alone does not dictate the theoretical or practical utility of a study. In fact, cross-sectional studies can provide an early foundation for unexplored areas of research that pave the way for future research using more intensive methodological procedures.
As such, we are hoping that these points will be considered when evaluating our paper for publication in Behavioral Sciences.